# Health impact and cost-effectiveness analysis of gender-neutral versus female-only 9-valent human papillomavirus vaccination in Taiwan

Hung-Hsueh Chou[1,2], Pen-Yuan Chu[3], Ying-Hui Wu[4], Casey Feng[4], Wei Wang[5], Isaya Sukarom[6]*, Andrew Pavelyev[5]

1 Chang Gung Memorial Hospital, Taoyuan, Taiwan, 2 National Tsing Hua University, School of Medicine, Hsinchu, Taiwan, 3 Taipei Veterans General Hospital, Taipei, Taiwan, 4 MSD Taiwan, Taipei, Taiwan, 5 Merck & Co., Inc., Rahway, New Jersey, United States of America, 6 MSD Thailand, Bangkok, Thailand

* isaya.sukarom@merck.com

## Abstract

The prevalence of human papillomavirus (HPV)-related diseases in women has declined in countries introducing HPV vaccinations for girls but remains high among men. The objective of this study was to compare the health impact and cost-effectiveness of two HPV vaccination strategies in the pre-adolescent Taiwanese population: nonavalent (9vHPV) gender-neutral vaccination (GNV), and 9vHPV female-only vaccination (FOV). A previously validated dynamic transmission model was adapted to the Taiwanese setting. The model had a 100-year time horizon and assumed an 85% vaccination coverage rate for girls and a 50% rate for boys, lifelong duration of vaccine protection, herd immunity, and a discount rate of 3% for costs and quality-adjusted life-years (QALYs). Costs of vaccination and HPV-related disease (in 2015−2016 new Taiwan dollars [NTD]), QALYs, and incremental cost-effectiveness ratios (ICERs) were estimated. Compared to 9vHPV FOV, GNV prevented 572 additional cases of cervical cancer (a 1.0% decrease) and 57,691 cases of genital warts (−12.6%) in women. In men, 26 additional cases of penile cancer were avoided (−6.6%), as well as 179,207 cases of genital warts (−26.7%), 4,955 cases of head and neck cancer (−9.0%), and 3,880 cases of recurrent respiratory papillomatosis (−15.8%), by use of the 9vHPV GNV strategy versus FOV. Use of 9vHPV GNV instead of FOV would prevent 229 deaths from cervical cancer in women (a 0.6% decrease) and 3,398 deaths from head and neck cancer in men (−8.3%) over 100 years. The 9vHPV GNV strategy resulted in a savings of NTD 1,574,288,155 (1.9%) in disease management costs compared to the 9vHPV FOV strategy and was predicted to be cost-effective, with an ICER of NTD 606,210/QALY. Compared to a 9vHPV FOV strategy, a 9vHPV GNV strategy for 13-year-old girls and boys would result in incremental public health and economic benefits and would be cost-effective in Taiwan.

**Data availability statement:** All relevant data are within the manuscript and its Supporting Information files.

**Funding:** This study was funded by Merck Sharp & Dohme LLC, a subsidiary of Merck & Co., Inc., Rahway, NJ, USA. The funder provided support in the form of salaries for authors YW, CF, WW, IS, and AP, but did not have any additional role in the study design, data collection and analysis, decision to publish, or preparation of the manuscript. The specific roles of these authors are articulated in the 'author contributions' section.

**Competing interests:** Hung-Hsueh Chou received honorarium for lectures/presentations/educational events from Merck Sharp & Dohme LLC, a subsidiary of Merck & Co., Inc., Rahway, NJ, USA, Takeda Pharmaceuticals, and AstraZeneca Pen-Yuan Chu has no conflict of interest to report. Ying-hui Wu and Casey Feng are employees of MSD Taiwan, Taipei, Taiwan. Isaya Sukarom is an employee of MSD Thailand, Bangkok, Thailand. Wei Wang and Andrew Pavelyev are employees of Merck Sharp & Dohme LLC, a subsidiary of Merck & Co., Inc., Rahway, NJ, USA at the time of the study. This does not alter our adherence to PLOS ONE policies on sharing data and materials.

## Introduction

Human papillomavirus (HPV) can cause cervical, vulvar, and vaginal cancers and precancers in women; penile cancers and precancers in men; and head and neck cancers, anal cancer, and genital warts in both genders [1,2]. Globally, the most prevalent HPV-related cancer in women is cervical cancer, while the most prevalent HPV-related cancer in men is head and neck cancer [3,4]. In 2017, there were 1,418 new cases of cervical cancer in Taiwan, and the mortality rate of cervical cancer was 5.49 per 100,000 [5]. That same year, the incidence of head and neck cancer in Taiwan was 33.6 per 100,000 [5,6].

Different genotypes of HPV are associated with genital neoplasia, various cancers, and genital warts [7–10]. Globally, HPV types 6 and 11 are known to be responsible for the majority of genital wart cases [11]. The leading genotypes in cervical cancer among Taiwanese women are HPV 16, 18, 33, 52, and 58 [12]. The leading HPV type identified in oropharyngeal cancers in Taiwan is HPV 16 [13]. A bivalent HPV (2vHPV) vaccine became available in 2007 that targets HPV types 16 and 18, followed by the release of a nonavalent HPV (9vHPV) vaccine in 2015 that targets types 6, 11, 31, 33, 45, 52, and 58 (in addition to 16 and 18) [14]. (A quadrivalent vaccine targeting types 6, 11, 16, and 18 was also released in 2006, but it exited the market in 2020.) Both the bivalent and nonavalent vaccines are available in Taiwan, and since December 2018, the Taiwanese government had provided a two-dose regimen of the 2vHPV vaccine free of charge to 13-year-old girls as part of the national immunization program. Starting in 2022, the 9vHPV vaccine has been used in place of the 2vHPV vaccine in this program.

While a declining trend in the incidence/prevalence of HPV infection and HPV-related diseases has been observed since the introduction of HPV vaccinations for girls in many countries [15,16], the burden of HPV-related diseases among men, particularly oropharyngeal cancer and anal cancer, remains high [17,18]. Given that men have a low rate of seroconversion following natural infection with HPV types 6, 11, 16, and 18 [19], leaving them at risk of subsequent infections throughout their lifetime [20], there is a strong rationale to include males in HPV immunization programs.

The goal of this study was to assess the health and economic impact and cost-effectiveness of implementing gender-neutral vaccination (GNV) with the 9vHPV vaccine in Taiwan. A GNV strategy was compared to the current national program of female-only vaccination (FOV) with the 9vHPV vaccine.

## Methods

### Model design

A previously validated dynamic transmission model simulating the natural history of HPV infections (Fig 1) and estimating the cost associated with HPV-related diseases [21,22] was adapted to the Taiwanese setting to compare two national immunization strategies: 9vHPV GNV and 9vHPV FOV from the healthcare payer perspective. Individuals enter the model at birth at a gender-specific and sexual activity-specific rate and move between successive age groups at an age- and gender-specific rate

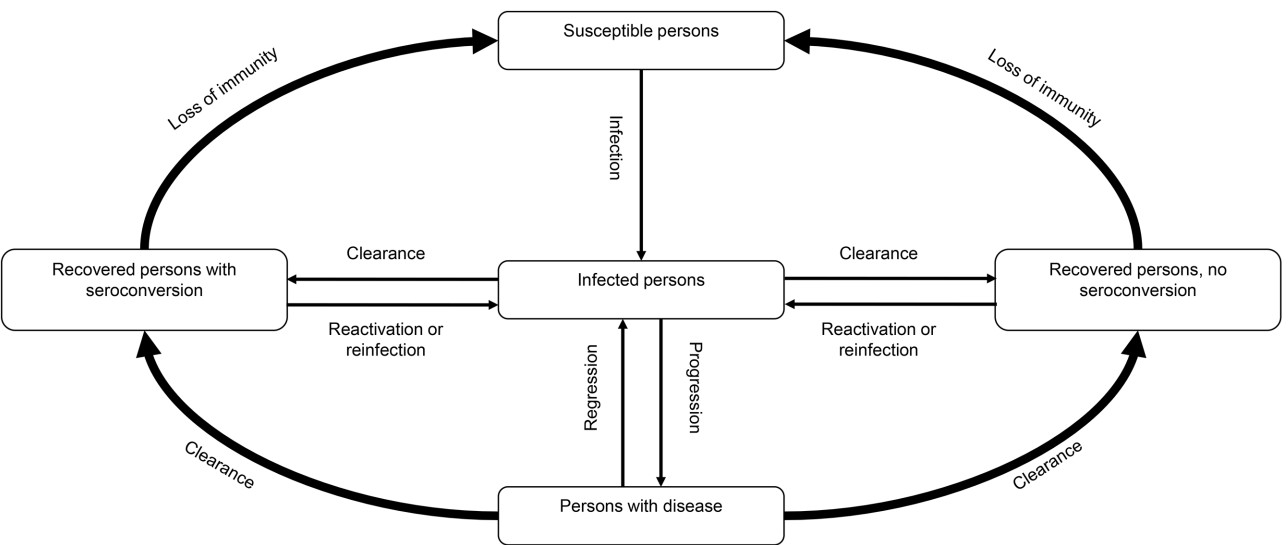

**Fig 1. A simplified schematic diagram of compartments.** The schematic diagram was adapted from Elbasha et al. (2010) [21].

per year. Individuals exit the model when they die, in accordance with known or estimated age- and disease-dependent death rates. HPV transmission and the occurrence of cervical intraepithelial neoplasia (CIN), cervical cancer, external genital warts, and other HPV-related diseases were simulated in the model. Progression of individuals from acquisition of an infection to disease was modeled using a natural history structure similar to that described in previous models for HPV 16/18 [23]. Attribution of disease to specific HPV types was performed based on published findings [7,13,24–31] (S1 Table).

The target population for 9vHPV GNV was 12- and 13-year-old boys and girls, and that for 9vHPV FOV was girls aged 12–13 years. The base case scenario assumed an 85% vaccination coverage rate for girls [32] and a 50% coverage rate for boys [33], as well as lifelong duration of vaccine protection, herd immunity, and a discount rate of 3% for costs and health outcomes. The time horizon was 100 years. Sensitivity analyses were conducted in which the vaccination coverage rate in boys ranged from 40% to 80% [34], vaccine price and treatment costs were varied +/- 20%, an alternate duration of protection (20 years instead of lifelong) was tested, and different discount rate scenarios (no discount and 5% discount) were considered.

## Model inputs

The model comprised three modules: (i) demographic and behavioral variables, (ii) epidemiologic variables, and (iii) economic variables.

Demographic variables were obtained from Taiwan's Ministry of the Interior [35] and included a population count of 11,719,580 (49.72%) males and 11,851,647 (50.28%) females as of 2017 (S2 Table). Age-specific annual all-cause mortality rates for the general population were also from the Ministry of the Interior and were used to determine when individuals exited the model (i.e., by death from non-HPV-related causes). Sexual activity parameters consisted of the proportions of the population with low (0–1 sexual partners per year), medium (2–4 sexual partners per year), and high (5 or more sexual partners per year) sexual activity (S2 Table). These sexual activity levels corresponded to the level of risk of HPV infection.

Epidemiologic variables included data on screening and treatment patterns (S3 Table). Screening rates for cervical cancer in females were obtained from the Cervical Cancer Screening Registry's 2018 Annual Report [36] and were expressed as the percentage of females receiving a follow-up screening test after an abnormal PAP smear (88%), the percentage of females receiving gynecological cancer screening tests at least once in their lifetime (88%), and the age-specific percentage of females receiving cervical cancer screening tests in the past year, as well as the specificity of PAP screening results (98.6% [37]; S3 Table).

Treatment patterns were obtained from the National Health Insurance Research Database (NHIRD) ambulatory care and inpatient claims for 2015–2016 [38] (S3 Table). These inputs included age-specific rates of hysterectomy.

HPV vaccination rates were obtained from Taiwan's Ministry of Health, which reported annual vaccination rates in females aged 13–14 of 73% in 2019 and 87% in 2020 [32]. The expected vaccination rate for males in the base case was 50% [33]. The prophylactic efficacy, or degree of protection, offered by the vaccine was based on clinical trial data [39–44]. Natural history inputs were obtained from the Taiwan Cancer Registry [45] and consisted of age- and sex-specific mortality rates for each HPV-related disease.

Economic variables consisted of the costs for vaccination, screening and diagnosis, and episodes of care in 2015–2016 new Taiwan dollars (NTD), as well as health utility values. The cost of vaccination was the cost of the vaccine itself (NTD 1,880 per dose) plus the cost of administration (NTD 100 per dose), as reported by the Health Promotion Administration (S4 Table) [46]. The Taiwan-specific costs for screening and diagnosis of cervical and vaginal cancers were obtained from the NHIRD [38] and included office visits (NTD 230), colposcopy (NTD 605), and biopsy (NTD 2,171; S4 Table). Costs for episodes of care were obtained from 2014–2015 NHIRD inpatient and ambulatory claims [38] and included CIN and carcinoma in situ for females, along with location-specific (i.e., local, regional, distant) cervical, vaginal, vulvar, anal, and head and neck cancers; genital warts; and recurrent respiratory papillomatosis (S4 Table). Costs for episodes of care in males included location-specific penile, anal, and head and neck cancer, along with genital warts and recurrent respiratory papillomatosis. The health utility values for the Taiwanese general population and for those with HPV-related diseases were derived from Elbasha *et al.* (2010) [21].

## Model outputs

HPV-associated health outcomes were estimated for each vaccination strategy (9vHPV FOV and 9vHPV GNV) and included the incidence (i.e., case counts) of cervical lesions (CIN 1/2/3); cervical, vaginal, vulvar, penile, anal, and head and neck cancers; as well as genital warts and recurrent respiratory papillomatosis. Mortality rates (i.e., cumulative deaths) were estimated for each cancer type as well as for recurrent respiratory papillomatosis. Costs and quality-adjusted life-years (QALYs) associated with HPV-related diseases were calculated for each vaccination strategy.

## Statistical analysis

Calibration was performed to ensure consistency between epidemiologic outputs and real-world observations (S5 Table). The targets of the calibration were publicly available data on incidence and mortality rates (from NHIRD 2015–2016 [38] and the 2019 Causes of Death database, Ministry of Health) of HPV-related diseases in Taiwan, adjusted for the proportion of incidence and mortality attributable to HPV types targeted by the nonavalent vaccine. All parameters were initially set to the values from the US model [22] and then adjusted until convergence was achieved between the target data and model output for the cervical cancer incidence rate and the cervical cancer mortality rate. Subsequently, the per sexual partnership transmission probability of multiple HPV types was adjusted to achieve the correct incidence and mortality.

Reductions in incidence and mortality rates were expressed as cases and deaths avoided under the 9vHPV GNV strategy versus the 9vHPV FOV strategy. Cost differences with 9vHPV GNV versus 9vHPV FOV were used to calculate the incremental cost-effectiveness ratio (ICER; incremental cost difference divided by the incremental QALY difference).

The threshold for determining cost-effectiveness was NTD 924,796/QALY, reflecting the per capita gross domestic product (GDP) in Taiwan [47] and the maximum willingness of decision-makers to pay for an additional QALY.

## Results

### Health impact of gender-neutral vaccination

Compared to 9vHPV FOV, the 9vHPV GNV strategy prevented more cases of HPV-related diseases over a 100-year period (Table 1). In women, 9vHPV GNV prevented 572 additional cases of cervical cancer and 6,463 additional cases of CIN 1/2/3. The largest reduction in female case counts was for genital warts: 57,691 cases avoided, a 12.6% reduction in incidence. In men, 26 additional cases of penile cancer were avoided by use of the 9vHPV GNV strategy compared to 9vHPV FOV, a 6.6% reduction in cumulative incidence. Men also avoided 179,207 additional cases of genital warts (a 26.7% reduction), 4,955 additional cases of head and neck cancer (a 9.0% reduction), and 3,880 additional cases of recurrent respiratory papillomatosis (a 15.8% reduction) with 9vHPV GNV.

Compared to the 9vHPV FOV strategy, 9vHPV GNV prevented 229 additional deaths from cervical cancer (a 0.6% reduction) in women (Table 1). The largest differential reduction in women's mortality was for recurrent respiratory papillomatosis (10.7%; n = 118 deaths). The effects on mortality were greater in men, where the 9vHPV GNV strategy prevented 3,398 additional deaths from head and neck cancer (an 8.3% reduction) and caused a 14.1% reduction in deaths from recurrent respiratory papillomatosis (n = 174) and a 6.1% reduction in deaths from penile cancer (n = 15) compared to the 9vHPV FOV strategy (Table 1).

**Table 1. Cases and deaths avoided and cumulative reduction in HPV-related disease incidence and mortality with 9vHPV gender-neutral vaccination vs. 9vHPV female-only vaccination over 100 years.**

|  | Number of cases avoided (% reduction) | Number of deaths avoided (% reduction) |
|---|---|---|
| Cervical cancer | 572 (1.0) | 229 (0.6) |
| CIN 1 | 3,694 (0.7) | |
| CIN 2/3 | 2,769 (0.6) | |
| Vaginal cancer | 5 (1.4) | 2 (1.3) |
| Vulvar cancer | 5 (1.5) | 2 (1.4) |
| Genital warts | | |
| Females | 57,691 (12.6) | |
| Males | 179,207 (26.7) | |
| HPV 6/11-related CIN 1 | 30,309 (11.7) | |
| Anal cancer | | |
| Females | 89 (3.5) | 32 (3.3) |
| Males | 251 (11.3) | 152 (10.9) |
| Penile cancer | 26 (6.6) | 15 (6.1) |
| Head and neck cancers | | |
| Females | 175 (3.1) | 78 (2.8) |
| Males | 4,955 (9.0) | 3,398 (8.3) |
| Recurrent respiratory papillomatosis | | |
| Females | 2,616 (12.1) | 118 (10.7) |
| Males | 3,880 (15.8) | 174 (14.1) |

CIN, cervical intraepithelial neoplasia; HPV, human papillomavirus.

## Economic impact of gender-neutral vaccination

The 9vHPV GNV strategy resulted in savings of NTD 1,574,288,155 in disease management costs compared to the 9vHPV FOV strategy (a 1.9% reduction). As shown in Fig 2, these reductions were attributable to diseases associated with HPV 16/18 (43.0%), HPV 6/11 (46.4%), and HPV 31/33/45/52/58 (10.6%). Cost savings peaked after about 55 years of implementation.

In the base case scenario, the 9vHPV GNV strategy was predicted to be cost-effective per our definition, with an ICER of NTD 606,210/QALY (Table 2). This was based on an incremental cost increase of NTD 358.23 per person, which was offset by a gain of 0.00059 QALYs per person with the GNV strategy. The sensitivity analysis using a range of vaccination coverage rates in males (40% to 80%) showed that any of these vaccination rates would be cost-effective, with ICERs ranging from NTD 596,478/QALY to NTD 631,711/QALY (Table 2). Similarly, sensitivity analyses that varied vaccine price, treatment costs, duration of protection, and a no discount scenario showed that a GNV strategy remained cost-effective under these scenarios (Table 2).

## Discussion

In this study, we evaluated the potential health and economic impact of GNV with the 9vHPV vaccine compared with FOV in Taiwan. Using a validated dynamic transmission model, a GNV approach was projected to reduce HPV-related disease incidence, mortality, and costs compared with a FOV approach, and the ICER estimates were below the threshold for cost-effectiveness in Taiwan. A program of 9vHPV GNV was projected to reduce HPV-related diseases beyond cervical cancer, including CIN in women and penile, anal, and head and neck cancers in men. The incidence of genital warts was also projected to be greatly reduced in both sexes. Thus, implementing a national strategy of 9vHPV GNV is predicted not only to further reduce cervical cancer and other HPV-related diseases in women who already benefit from 2vHPV and 9vHPV vaccination, but also to usher in significant reductions in HPV-related diseases experienced by men.

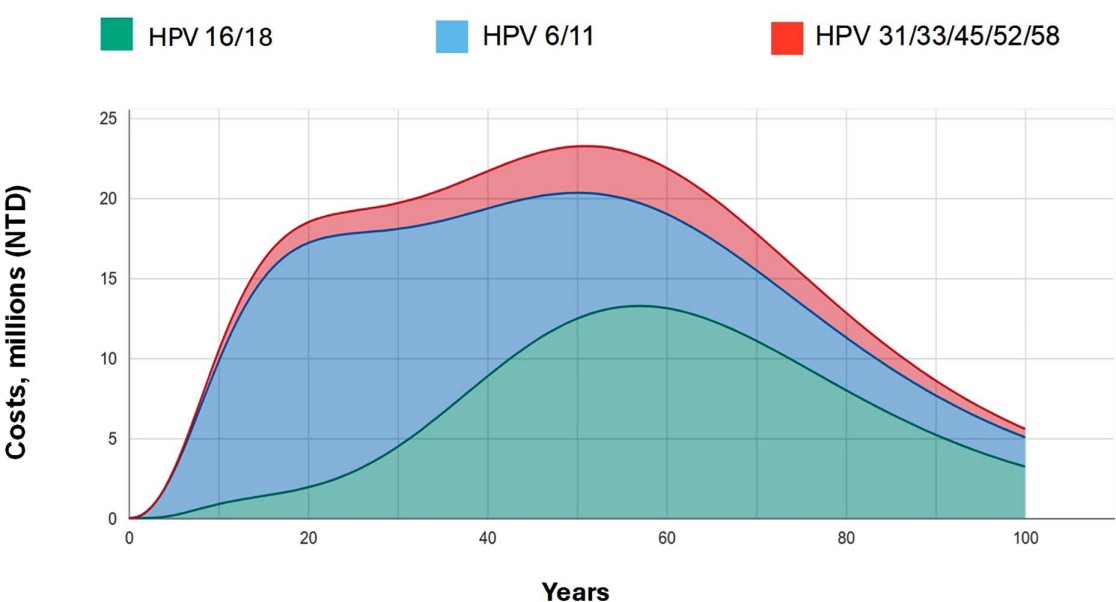

**Fig 2. HPV genotype-specific treatment costs avoided over 100 years with 9vHPV gender-neutral vaccination vs. 9vHPV female-only vaccination.** HPV, human papillomavirus.

**Table 2. Cost-effectiveness of the 9vHPV gender-neutral vaccination vs. 9vHPV female-only vaccination.**

| | VCR (female/male) | Cost/person (NTD) | QALY/person (years) | Incremental cost/person (NTD) | Incremental QALYs/person (years) | ICER (NTD/QALY) |
|---|---|---|---|---|---|---|
| **Base case analysis** | | | | | | |
| 9vHPV female-only vaccination | 85%/0% | 4,990.08 | 27.41571 | | | |
| 9vHPV gender-neutral vaccination | 85%/50% | 5,348.31 | 27.41630 | 358.23 | 0.00059 | 606,210 |
| **Sensitivity analyses** | | | | | | |
| **VCR** | | | | | | |
| 9vHPV gender-neutral vaccination | 85%/40% | 5,275.85 | 27.41619 | 285.77 | 0.00048 | 596,478 |
| 9vHPV gender-neutral vaccination | 85%/60% | 5,421.13 | 27.41641 | 431.05 | 0.00070 | 615,451 |
| 9vHPV gender-neutral vaccination | 85%/70% | 5,494.36 | 27.41652 | 504.28 | 0.00081 | 624,057 |
| 9vHPV gender-neutral vaccination | 85%/80% | 5,568.07 | 27.41663 | 577.99 | 0.00091 | 631,711 |
| **Vaccine price** | | | | | | |
| +20% | | | | | | |
| 9vHPV female-only vaccination | 85%/0% | 5,117.12 | 27.41571 | | | |
| 9vHPV gender-neutral vaccination | 85%/50% | 5,556.07 | 27.41630 | 438.95 | 0.00059 | 742,821 |
| −20% | | | | | | |
| 9vHPV female-only vaccination | 85%/0% | 4,863.04 | 27.41571 | | | |
| 9vHPV gender-neutral vaccination | 85%/50% | 5,140.54 | 27.41630 | 277.50 | 0.00059 | 469,599 |
| **Treatment costs** | | | | | | |
| +20% | | | | | | |
| 9vHPV female-only vaccination | 85%/0% | 5,697.81 | 27.41571 | | | |
| 9vHPV gender-neutral vaccination | 85%/50% | 6,042.69 | 27.41630 | 344.87 | 0.00059 | 583,616 |
| −20% | | | | | | |
| 9vHPV female-only vaccination | 85%/0% | 4,282.16 | 27.41571 | | | |
| 9vHPV gender-neutral vaccination | 85%/50% | 4,653.74 | 27.41630 | 371.58 | 0.00059 | 628,813 |
| **Duration of protection** | | | | | | |
| 20 years | | | | | | |
| 9vHPV female-only vaccination | 85%/0% | 5,583.05 | 27.41365 | | | |
| 9vHPV gender-neutral vaccination | 85%/50% | 5,937.36 | 27.41427 | 354.31 | 0.00062 | 568,385 |
| **Discount** | | | | | | |
| No discount | | | | | | |
| 9vHPV female-only vaccination | 85%/0% | 11,997.70 | 85.51721 | | | |
| 9vHPV gender-neutral vaccination | 85%/50% | 12,918.80 | 85.52118 | 921.13 | 0.00397 | 232,314 |
| 5% discount | | | | | | |
| 9vHPV female-only vaccination | 85%/0% | 3,505.09 | 17.38564 | | | |
| 9vHPV gender-neutral vaccination | 85%/50% | 3,753.33 | 17.38587 | 248.24 | 0.00022 | 1,126,248 |

HPV, human papillomavirus; ICER, incremental cost-effectiveness ratio; NTD, new Taiwan dollars; QALY, quality-adjusted life year; VCR, vaccination coverage rate.

Several previous studies have used static Markov models to establish the cost-effectiveness of the bivalent, quadrivalent, and nonavalent HPV vaccines in Taiwan [48–50], but health and economic outcomes in these studies were assessed primarily in terms of cervical cancer prevention, and an inherent limitation of Markov models is that they cannot account for herd immunity. Two other studies have evaluated HPV vaccination in Taiwan using dynamic transmission models [51,52]. Dasbach *et al.* assessed the epidemiological and economic effects of quadrivalent HPV vaccination in conjunction with cervical cancer screening in Taiwan using scenarios with and without a catch-up program (i.e., vaccination of females

up to age 24) [52]. They found that, compared to no vaccination, both scenarios reduced cervical cancer by 91%, CIN2/3 by 90%, and genital warts by 94%, but that the catch-up scenario achieved these outcomes sooner within a 100-year time horizon. More recently, Chou *et al.* evaluated the impact of nonavalent HPV vaccination for 13–14-year-old females in Taiwan compared to the bivalent vaccine [51]. This study showed that nonavalent vaccination would prevent an additional 15,951 cases of cervical cancer, reduce cervical cancer mortality by 18.0%, and prevent an additional 1,115,317 cases of HPV6/11-related genital warts compared to bivalent vaccination, producing a cost savings of NTD 762,840,226 over 100 years.

All of the aforementioned studies have one thing in common: they focused exclusively on outcomes in women. The primary strength of the current study is that it demonstrates the benefits of GNV for both women and men. Our model projected decreases of ~10% or more in anogenital cancers, genital warts, and recurrent respiratory papillomatosis in men. In contrast, some of the effects we observed in women were nominal (e.g., the maximum reduction in female cancer incidence was 3.5% for anal cancer); this is because we compared 9vHPV GNV against a baseline of 9vHPV FOV, where most of the benefits in women are achieved already (as demonstrated by Chou *et al.* (2022) [51]). This pattern is consistent with previous studies from other countries as well [53–56]. In Italy, GNV with the 9vHPV vaccine produced additional decreases of 4% in cervical cancer incidence, 4% in CIN1 incidence, 4% in CIN2 + incidence, and 7% in genital warts incidence among females compared to 9vHPV FOV [55]. In Spain, the inclusion of boys in the 9vHPV national immunization program was predicted to result in reductions of 29.2% and 44.3% in the incidence of genital warts in females and males, respectively, and 3.6–6.4% in the incidence of cervical pathology in females [53].

Previous reports have noted the decreased health and economic benefits of GNV in the presence of high female vaccination rates [56–58]. This renders our findings on reduced incidence and mortality in males in the context of an 85% female vaccination coverage rate even more noteworthy. Such findings are consistent with the demonstrated efficacy of HPV vaccination against HPV-related disease in men [59–61]. Furthermore, our sensitivity analyses suggest that GNV would be cost-effective at a variety of vaccination coverage rates in males.

In Taiwan, the national immunization program recommendation under consideration is for full-series, two-dose, on-label use of the HPV vaccine for the cohorts included in this study. As such, our analyses focused specifically on the two-dose regimen and did not consider one-dose vaccination strategies, as they fall outside the scope of Taiwan's current vaccination strategy. Although emerging evidence suggests the potential 10-year effectiveness of single-dose HPV vaccination, there remains uncertainty regarding its public health impact, namely on cancers and diseases prevented. For example, a cost-effectiveness analysis by Daniels *et al.* (2022) found a significant increase in HPV-related cancer cases over a 100-year period when comparing a one-dose versus two-dose vaccination program in the United Kingdom [62]. Furthermore, the one-dose vaccination program was also found to be less cost-effective than the two-dose vaccination program [62]. Similar conclusions have also been observed in the context of low-income countries such as Indonesia, in which two-dose HPV vaccination programs were found to be more cost-effective compared to one-dose programs across various scenarios [63].

Our results supporting the two-dose GNV strategy with 9vHPV further align with findings from Hong Kong. In Hong Kong, compared to routine FOV with the 9vHPV vaccine, GNV with the 9vHPV vaccine was predicted to result in a 4.5% reduction in cervical cancer cases, 7.4% reduction in CIN 1 cases, and a 6.9% reduction in CIN 2/3 cases [64]. Further, GNV was predicted to result in reductions in both female and male anal cancer cases (4.6%/14.6%, respectively), head and neck cancer cases (4.8%/16.2%, respectively), genital warts (33.2%/51.1%), and 24.9% of penile cancer cases over 100 years [64].

In the present study, the 9vHPV GNV strategy resulted in savings of NTD 1,574,288,155 in disease management costs compared to the 9vHPV FOV strategy. These cost reductions were attributable to diseases associated with HPV 16/18 (43.0%), HPV 6/11 (46.4%), and HPV 31/33/45/52/58 (10.6%). Notably, the incremental cost of implementing

GNV—estimated at NTD 358.23 per person in the base case—can be contextualized as a percentage of the current cervical cancer screening budget (NTD 941 million) or the broader cancer prevention budget (NTD 3.398 billion), offering policymakers a clearer picture of the investment-to-benefit ratio [65]. These cost-reduction results align with those found in a modeling exercise in Hong Kong, where cost reductions attributable to HPV-related diseases were 52.1% for HPV16/18, 41.4% for HPV6/11, and 6.6% for HPV 31/33/45/52/58 [64]. These findings may be particularly valuable for other low- and middle-income countries considering the adoption of a 9vHPV GNV strategy where budgetary constraints often limit access to broader HPV vaccination programs, and the burden of HPV-related diseases remains significant.

Our study, like all modeling studies, is subject to various limitations. The model did not include all potential health benefits or economic benefits of HPV vaccination. For example, the effect of catch-up vaccination strategies was not assessed, and disease transmission among males who have sex with other males was not included. Indirect costs associated with HPV-related diseases, such as lost productivity, were not taken into consideration. These factors may have resulted in the underestimation of the added benefit of male vaccination compared with FOV. The model also did not account for potential changes to cervical cancer screening methods that may develop over the course of 100 years. The model was calibrated using age-standardized incidence rates. Calibrating model outputs to the age-standardized rate of cancer incidence instead of using age-specific incidence rates may simplify the analysis, but may also lead to a loss of granularity of the data, thereby affecting the projected incidence, associated healthcare cost, and health benefits across different age cohorts over time. For example, age-standardized rates allow for a more uniform comparison across different populations and time periods by removing the effects of differing age distributions. This can lead to a more accurate understanding of trends in cancer incidence that are not influenced by demographic shifts. However, it could also mask important trends in specific age groups that may indicate increasing or decreasing risks associated with certain cancers, which in turn may under or overestimate healthcare costs. That is, if an age-specific rate indicates a rising incidence in older adults, cost projections tied to treatment and care for this demographic could be higher than if only age-standardized rates are considered. Conversely, focusing solely on age-standardized rates might overlook increasing needs for younger populations facing rising incidences of certain cancers. Similar implications can be delineated for health benefit assessments, policy, and resource allocations. Age-standardized rates might obscure these nuances, especially if certain age cohorts are increasingly diagnosed with specific types of cancer. Finally, cost-effectiveness estimates are sensitive to vaccination coverage rates and vaccine costs, among other variables, and the high female coverage rate assumed in the current analysis may underestimate the effects observed in other settings with lower female coverage rates. Also, while the use of Taiwan-specific inputs strengthens the reliability of the projections for Taiwan, it means that our interpretations are generalizable only to settings with a comparable medical and economic context.

## Conclusions

In conclusion, dynamic transmission modeling showed that, compared to a 9vHPV FOV strategy, a 9vHPV GNV strategy for 13-year-old girls and boys in Taiwan would result in incremental public health and economic benefits and would be cost-effective relative to the per capita GDP of Taiwan. A 9vHPV GNV approach would not only help accelerate the timeline of achieving cervical cancer elimination but also reduce the incidence of HPV-related non-cervical cancers such as anal and head and neck cancer.

## Supporting information

**S1 Table. Attribution of disease to specific HPV types.**
(DOCX)

**S2 Table. Inputs for the demographic module: population counts, annual mortality, and sexual behavior.**
(DOCX)

**S3 Table. Epidemiologic input parameters: screening and treatment patterns.**
(DOCX)

**S4 Table. Economic input parameters: costs of vaccination, screening, and treatmen.**
(DOCX)

**S5 Table. Calibration results.**
(DOCX)

## Acknowledgments

The authors thank Dr. Alhaji Cherif for discussion of the project and manuscript. The authors also thank Melissa Stauffer, PhD, in collaboration with ScribCo, for medical writing assistance.

## Author contributions

**Conceptualization:** Hung-Hsueh Chou, Pen-Yuan Chu, Ying-hui Wu, Casey Feng, Wei Wang, Isaya Sukarom, Andrew Pavelyev.

**Data curation:** Hung-Hsueh Chou, Pen-Yuan Chu, Ying-hui Wu, Casey Feng, Wei Wang, Isaya Sukarom, Andrew Pavelyev.

**Formal analysis:** Andrew Pavelyev.

**Methodology:** Ying-hui Wu, Casey Feng, Isaya Sukarom, Andrew Pavelyev.

**Supervision:** Hung-Hsueh Chou, Pen-Yuan Chu, Ying-hui Wu, Casey Feng, Wei Wang, Isaya Sukarom, Andrew Pavelyev.

**Validation:** Hung-Hsueh Chou, Pen-Yuan Chu, Ying-hui Wu, Casey Feng, Wei Wang, Isaya Sukarom, Andrew Pavelyev.

**Writing – original draft:** Hung-Hsueh Chou, Pen-Yuan Chu, Ying-hui Wu, Casey Feng, Wei Wang, Isaya Sukarom, Andrew Pavelyev.

**Writing – review & editing:** Hung-Hsueh Chou, Pen-Yuan Chu, Ying-hui Wu, Casey Feng, Wei Wang, Isaya Sukarom, Andrew Pavelyev.

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
