## [Decision Letter · Decision Letter 0]

16 Sep 2024

PONE-D-24-13297Health impact and cost-effectiveness analysis of gender-neutral versus female-only 9-valent human papillomavirus vaccination in TaiwanPLOS ONE

Dear Dr. Sukarom,

Thank you for submitting your manuscript to PLOS ONE. After careful consideration, we feel that it has merit but does not fully meet PLOS ONE’s publication criteria as it currently stands. Therefore, we invite you to submit a revised version of the manuscript that addresses the points raised during the review process.

The main limitation of the report is that the model used in the study assumes a lifelong duration of vaccine protection and herd immunity which may be unrealistic and unsubstantiated by available data. This assumption potentially overestimates the long-term benefits of vaccination if vaccine-induced immunity wanes over time or if coverage rates fall below expectations.

Authors are invited to comment convincingly on the long time horizon of 100 years which introduces uncertainty due to potential changes in healthcare policies, and screening programmes over time, which are not accounted for in the model. For example, the model is completely silent about emerging evidence on the 10-year effectiveness of single-dose HPV vaccination and the potential implications for vaccination programmes.

From a research ethics perspective, and in line with Journal policy, authors need to do more to enhance the reproducibility of their findings (eg. Through open-access disclosure of data files) and generalizability to other settings (eg. Currencies, vaccine coverage flexibility in line with WHO cervical cancer elimination agenda). This follows logically from the observation that while the model is adapted to the Taiwanese setting, using Taiwan-specific data for disease incidence and mortality, certain inputs are based on studies from other countries.

We look forward to receiving your revised manuscript.

Kind regards,

Emmanuel Timmy Donkoh, PhD

Academic Editor

PLOS ONE

Journal Requirements:

https://pubmed.ncbi.nlm.nih.gov/37183965/

https://linkinghub.elsevier.com/retrieve/pii/S2212109922001315

https://www.eurogin.com/content/dam/markets/aest/eurogin/pdfs/2022/2022-Abstracts-MCP.pdf

https://pubmed.ncbi.nlm.nih.gov/37183965/

In your revision ensure you cite all your sources (including your own works), and quote or rephrase any duplicated text outside the methods section. Further consideration is dependent on these concerns being addressed.

4. We note that the grant information you provided in the ‘Funding Information’ and ‘Financial Disclosure’ sections do not match. When you resubmit, please ensure that you provide the correct grant numbers for the awards you received for your study in the ‘Funding Information’ section.

5.  Thank you for stating the following financial disclosure: “This study was funded by Merck Sharp & Dohme LLC, a subsidiary of Merck & Co., Inc., Rahway, NJ, USA”

6. Thank you for stating the following in the Competing Interests section: “I have read the journal's policy and the authors of this manuscript have the following competing interests:

Hung-Hsueh Chou received honorarium for lectures/presentations/educational events from Merck Sharp & Dohme LLC, a subsidiary of Merck & Co., Inc., Rahway, NJ, USA, Takeda Pharmaceuticals, and AstraZeneca

Pen-Yuan Chu has no conflict of interest to report.

Ying-hui Wu and Casey Feng are employees of MSD Taiwan, Taipei, Taiwan.

Isaya Sukarom is an employee of MSD Thailand, Bangkok, Thailand.

Wei Wang and Andrew Pavelyev are employees of Merck Sharp & Dohme LLC, a subsidiary of Merck & Co., Inc., Rahway, NJ, USA at the time of the study.”

We note that one or more of the authors are employed by a commercial company: MSD Taiwan, MSD Thailand and Merck Sharp & Dohme LLC

a. Please provide an amended Funding Statement declaring this commercial affiliation, as well as a statement regarding the Role of Funders in your study. If the funding organization did not play a role in the study design, data collection and analysis, decision to publish, or preparation of the manuscript and only provided financial support in the form of authors' salaries and/or research materials, please review your statements relating to the author contributions, and ensure you have specifically and accurately indicated the role(s) that these authors had in your study. You can update author roles in the Author Contributions section of the online submission form. Please also include the following statement within your amended Funding Statement. “The funder provided support in the form of salaries for authors [insert relevant initials], but did not have any additional role in the study design, data collection and analysis, decision to publish, or preparation of the manuscript. The specific roles of these authors are articulated in the ‘author contributions’ section.” If your commercial affiliation did play a role in your study, please state and explain this role within your updated Funding Statement.

7.  We note that your Data Availability Statement is currently as follows: “All relevant data are within the manuscript and its Supporting Information files.”

Please confirm at this time whether or not your submission contains all raw data required to replicate the results of your study. Authors must share the “minimal data set” for their submission. PLOS defines the minimal data set to consist of the data required to replicate all study findings reported in the article, as well as related metadata and methods (https://journals.plos.org/plosone/s/data-availability#loc-minimal-data-set-definition). For example, authors should submit the following data: - The values behind the means, standard deviations and other measures reported; - The values used to build graphs; - The points extracted from images for analysis. Authors do not need to submit their entire data set if only a portion of the data was used in the reported study. If your submission does not contain these data, please either upload them as Supporting Information files or deposit them to a stable, public repository and provide us with the relevant URLs, DOIs, or accession numbers. For a list of recommended repositories, please see https://journals.plos.org/plosone/s/recommended-repositories. If there are ethical or legal restrictions on sharing a de-identified data set, please explain them in detail (e.g., data contain potentially sensitive information, data are owned by a third-party organization, etc.) and who has imposed them (e.g., an ethics committee). Please also provide contact information for a data access committee, ethics committee, or other institutional body to which data requests may be sent. If data are owned by a third party, please indicate how others may request data access.

8.  We note that you have included the phrase “data not shown” in your manuscript. Unfortunately, this does not meet our data sharing requirements. PLOS does not permit references to inaccessible data. We require that authors provide all relevant data within the paper, Supporting Information files, or in an acceptable, public repository. Please add a citation to support this phrase or upload the data that corresponds with these findings to a stable repository (such as Figshare or Dryad) and provide and URLs, DOIs, or accession numbers that may be used to access these data. Or, if the data are not a core part of the research being presented in your study, we ask that you remove the phrase that refers to these data.

9. We note that there is identifying data in the Supporting Information Table S1 and S2. Due to the inclusion of these potentially identifying data, we have removed this file from your file inventory. Prior to sharing human research participant data, authors should consult with an ethics committee to ensure data are shared in accordance with participant consent and all applicable local laws.

-Ages more specific than whole numbers -Internet protocol (IP) address

 -Contact information such as phone number or email address –

Location data

Data that are not directly identifying may also be inappropriate to share, as in combination they can become identifying. For example, data collected from a small group of participants, vulnerable populations, or private groups should not be shared if they involve indirect identifiers (such as sex, ethnicity, location, etc.) that may risk the identification of study participants. Additional guidance on preparing raw data for publication can be found in our Data Policy (https://journals.plos.org/plosone/s/data-availability#loc-human-research-participant-data-and-other-sensitive-data) and in the following article: http://www.bmj.com/content/340/bmj.c181.long. Please remove or anonymize all personal information (<specific identifying information in file to be removed>), ensure that the data shared are in accordance with participant consent, and re-upload a fully anonymized data set. Please note that spreadsheet columns with personal information must be removed and not hidden as all hidden columns will appear in the published file.

Additional Editor Comments:

The main limitation of the report is that the model used in the study assumes a lifelong duration of vaccine protection and herd immunity which may be unrealistic and unsubstantiated by available data. This assumption potentially overestimates the long-term benefits of vaccination if vaccine-induced immunity wanes over time or if coverage rates fall below expectations.

Authors are invited to comment convincingly on the long time horizon of 100 years which introduces uncertainty due to potential changes in healthcare policies, and screening programmes over time, which are not accounted for in the model. For example the model is completely silent about emerging evidence on the 10-year effectiveness of single-dose HPV vaccination and the potential implications for vaccination programmes.

From a research ethics perspective, and in line with Journal policy, authors need to do more to enhance the reproducibility of their findings (eg. Through open-access disclosure of data files) and generalizability to other settings (eg. Currencies, vaccine coverage flexibility in line with WHO cervical cancer elimination agenda). This follows logically from the observation that while the model is adapted to the Taiwanese setting, using Taiwan-specific data for disease incidence and mortality, certain inputs are based on studies from other countries.

Reviewers' comments:

Reviewer's Responses to Questions

**Comments to the Author**

1. Is the manuscript technically sound, and do the data support the conclusions?

Reviewer #1: No

Reviewer #2: Yes

2. Has the statistical analysis been performed appropriately and rigorously? 

Reviewer #1: No

Reviewer #2: Yes

3. Have the authors made all data underlying the findings in their manuscript fully available?

Reviewer #1: Yes

Reviewer #2: No

4. Is the manuscript presented in an intelligible fashion and written in standard English?

Reviewer #1: Yes

Reviewer #2: Yes

5. Review Comments to the Author

Reviewer #1: This study aims to assess the health and economic impact and cost-effectiveness of implementing gender-neutral vaccination (GNV) with the 9vHPV vaccine in Taiwan. The manuscript, however, lacks some important information in the main text. Here are some points I would like the authors to consider to further highlight the contribution of the study.

1. What is your perspective on this economic evaluation study? I couldn't find it in the text.

2. What is the baseline year? Is it 2017?

3. What is the natural history of HPV in your dynamic transmission model? Please provide a diagram as Figure 1.

4. Could you please conduct both deterministic and probabilistic sensitivity analyses for your model?

5. Why did you set the time horizon to 100 years? Typically, the time horizon is set to match the average life expectancy.

6. Could you please combine Table 1 and Table 2? I don't think it is necessary to separate them.

7. This study is well worth discussing. While HPV brings limited benefits to the male population, it incurs significant costs. Please explain in the discussion section how the 9vHPV GNV strategy compares economically with other vaccine programs (e.g., hepatitis B and measles vaccines) in Taiwan.

Reviewer #2: The authors presented the analysis to estimate the impact of GNV on cervical cancer and other HPV-related diseases on both females and males. More details for adapting the model to the Taiwanese setting should be provided.

Model design

1. Lines 81-82, 89-91, 150-151, 154-156. The model used a previously developed model with adaption to the Taiwanese setting. “All parameters were initially set to the values from the US model and then adjusted to converge on the cervical cancer incidence rate and the cervical cancer mortality rate”.

More information should be provided in the Supplementary file to make the current submission a standalone study. In particular, please show the model fitting to the Taiwanese disease data and include the source of the Taiwanese disease data. Model parameters, such as progression rate, should be compared between the Taiwanese study and the original US study.

2. Lines 91-92. The attribution of specific HPV types should be presented explicitly with justification.

3. Lines 106-109. Young adults likely have more sexual partners when compared to adolescents and older adults. That is, if the mean number of sexual partners is pre-specified in each sexual activity categories, the proportion of population in the sexual activity categories should vary by age.

In Supplementary Table 1, The proportion of the sexual activity categories, with pre-specified mean number of sexual partners per year, were independent of ages. The authors please justify the assumption of this setting and explain how this affects the transmission dynamic. Sensitivity analysis that considers age-varying proportion could be provided.

4. Lines 114-115. “the percentage of females receiving gynecological cancer screening tests (88%)”. From Supplementary Table 2, the quoted percentage referred to the percentage that females received screening tests at least once in their lifetime and the screening test should be referred to cervical screening only. The proportion that females received a screening test in the past year was around 30%. Please clarify in the text.

5. Lines 128-129. The cost of vaccine itself was NTD1,880 per dose, or equivalently USD$60 per dose. This was lower than a recent Taiwanese HPV vaccination CEA study (quoted NTD2,300 per dose) (https://doi.org/10.1016/j.vhri.2022.06.006). Either price was lower than the one quoted by the Health Promotion Administration, Ministry of Health and Welfare (NTD3,000-7,000 per dose) (https://www.hpa.gov.tw/EngPages/Detail.aspx?nodeid=1752&pid=11888). The authors please justify this setting.

6. Provide the exchange rate of NTD to USD, GBP, or Euro.

6. PLOS authors have the option to publish the peer review history of their article (what does this mean?). If published, this will include your full peer review and any attached files.

Reviewer #1: No

Reviewer #2: No

---

## [Author Response · Author response to Decision Letter 1]

15 Jan 2025

January 15, 2025

Emmanuel Timmy Donkoh, PhD

Academic Editor

PLOS ONE

Dear Dr. Emmanuel Timmy Donkoh,

On behalf of my co-authors, I would like to thank you and the reviewers for your feedback on our submission entitled, “Health impact and cost-effectiveness analysis of gender-neutral versus female-only 9-valent human papillomavirus vaccination in Taiwan” (PONE-D-24-13297). Below please find our point-by-point responses to the comments raised during peer review.

Journal Requirements:

• Response: We confirm that the manuscript meets PLOS ONE’s style requirements.

https://pubmed.ncbi.nlm.nih.gov/37183965/

https://linkinghub.elsevier.com/retrieve/pii/S2212109922001315

https://www.eurogin.com/content/dam/markets/aest/eurogin/pdfs/2022/2022-Abstracts-MCP.pdf

https://pubmed.ncbi.nlm.nih.gov/37183965/

In your revision ensure you cite all your sources (including your own works), and quote or rephrase any duplicated text outside the methods section. Further consideration is dependent on these concerns being addressed.

• Response: The journal has identified two of our previous publications that use the same dynamic transmission model to estimate the of HPV vaccination in Hong Kong and Taiwan. Our Methods section does overlap with these two previous publications because we are utilizing the same model, however, both of these publications have been properly cited.

• Response: We confirm that there is no funding-related text in the manuscript. The complete funding statement is as follows: “The funder provided support in the form of salaries for authors YW, CF, WW, IS, and AP, but did not have any additional role in the study design, data collection and analysis, decision to publish, or preparation of the manuscript. The specific roles of these authors are articulated in the ‘author contributions’ section.”

4. We note that the grant information you provided in the ‘Funding Information’ and ‘Financial Disclosure’ sections do not match. When you resubmit, please ensure that you provide the correct grant numbers for the awards you received for your study in the ‘Funding Information’ section.

• Response: No grants were received. The funding information has been updated and now reads: “The funder provided support in the form of salaries for authors YW, CF, WW, IS, and AP, but did not have any additional role in the study design, data collection and analysis, decision to publish, or preparation of the manuscript. The specific roles of these authors are articulated in the ‘author contributions’ section.”

5. Thank you for stating the following financial disclosure: “This study was funded by Merck Sharp & Dohme LLC, a subsidiary of Merck & Co., Inc., Rahway, NJ, USA”

• Response: “The funder provided support in the form of salaries for authors YW, CF, WW, IS, and AP, but did not have any additional role in the study design, data collection and analysis, decision to publish, or preparation of the manuscript. The specific roles of these authors are articulated in the ‘author contributions’ section.”

6. Thank you for stating the following in the Competing Interests section: “I have read the journal's policy and the authors of this manuscript have the following competing interests:

Hung-Hsueh Chou received honorarium for lectures/presentations/educational events from Merck Sharp & Dohme LLC, a subsidiary of Merck & Co., Inc., Rahway, NJ, USA, Takeda Pharmaceuticals, and AstraZeneca

Pen-Yuan Chu has no conflict of interest to report.

Ying-hui Wu and Casey Feng are employees of MSD Taiwan, Taipei, Taiwan.

Isaya Sukarom is an employee of MSD Thailand, Bangkok, Thailand.

Wei Wang and Andrew Pavelyev are employees of Merck Sharp & Dohme LLC, a subsidiary of Merck & Co., Inc., Rahway, NJ, USA at the time of the study.”

We note that one or more of the authors are employed by a commercial company: MSD Taiwan, MSD Thailand and Merck Sharp & Dohme LLC

a. Please provide an amended Funding Statement declaring this commercial affiliation, as well as a statement regarding the Role of Funders in your study. If the funding organization did not play a role in the study design, data collection and analysis, decision to publish, or preparation of the manuscript and only provided financial support in the form of authors' salaries and/or research materials, please review your statements relating to the author contributions, and ensure you have specifically and accurately indicated the role(s) that these authors had in your study. You can update author roles in the Author Contributions section of the online submission form. Please also include the following statement within your amended Funding Statement. “The funder provided support in the form of salaries for authors [insert relevant initials], but did not have any additional role in the study design, data collection and analysis, decision to publish, or preparation of the manuscript. The specific roles of these authors are articulated in the ‘author contributions’ section.” If your commercial affiliation did play a role in your study, please state and explain this role within your updated Funding Statement.

• Response: The funding statement has been updated and now reads: “The funder provided support in the form of salaries for authors YW, CF, WW, IS, and AP, but did not have any additional role in the study design, data collection and analysis, decision to publish, or preparation of the manuscript. The specific roles of these authors are articulated in the ‘author contributions’ section.”

• Response: The Completing Interests Statement has been modified as requested and now reads:

“Hung-Hsueh Chou received honorarium for lectures/presentations/educational events from Merck Sharp & Dohme LLC, a subsidiary of Merck & Co., Inc., Rahway, NJ, USA, Takeda Pharmaceuticals, and AstraZeneca

Pen-Yuan Chu has no conflict of interest to report.

Ying-hui Wu and Casey Feng are employees of MSD Taiwan, Taipei, Taiwan.

Isaya Sukarom is an employee of MSD Thailand, Bangkok, Thailand.

Wei Wang and Andrew Pavelyev are employees of Merck Sharp & Dohme LLC, a subsidiary of Merck & Co., Inc., Rahway, NJ, USA at the time of the study.

This does not alter our adherence to PLOS ONE policies on sharing data and materials.”

7. We note that your Data Availability Statement is currently as follows: “All relevant data are within the manuscript and its Supporting Information files.”

Please confirm at this time whether or not your submission contains all raw data required to replicate the results of your study. Authors must share the “minimal data set” for their submission. PLOS defines the minimal data set to consist of the data required to replicate all study findings reported in the article, as well as related metadata and methods (https://journals.plos.org/plosone/s/data-availability#loc-minimal-data-set-definition). For example, authors should submit the following data: - The values behind the means, standard deviations and other measures reported; - The values used to build graphs; - The points extracted from images for analysis. Authors do not need to submit their entire data set if only a portion of the data was used in the reported study. If your submission does not contain these data, please either upload them as Supporting Information files or deposit them to a stable, public repository and provide us with the relevant URLs, DOIs, or accession numbers. For a list of recommended repositories, please see https://journals.plos.org/plosone/s/recommended-repositories. If there are ethical or legal restrictions on sharing a de-identified data set, please explain them in detail (e.g., data contain potentially sensitive information, data are owned by a third-party organization, etc.) and who has imposed them (e.g., an ethics committee). Please also provide contact information for a data access committee, ethics committee, or other institutional body to which data requests may be sent. If data are owned by a third party, please indicate how others may request data access.

• Response: We confirm that the manuscript includes all data needed to reproduce these results.

8. We note that you have included the phrase “data not shown” in your manuscript. Unfortunately, this does not meet our data sharing requirements. PLOS does not permit references to inaccessible data. We require that authors provide all relevant data within the paper, Supporting Information files, or in an acceptable, public repository. Please add a citation to support this phrase or upload the data that corresponds with these findings to a stable repository (such as Figshare or Dryad) and provide and URLs, DOIs, or accession numbers that may be used to access these data. Or, if the data are not a core part of the research being presented in your study, we ask that you remove the phrase that refers to these data.

• Response: The phrase “Data not shown” has been removed. The data is shown in the statement in question.

9. We note that there is identifying data in the Supporting Information Table S1 and S2. Due to the inclusion of these potentially identifying data, we have removed this file from your file inventory. Prior to sharing human research participant data, authors should consult with an ethics committee to ensure data are shared in accordance with participant consent and all applicable local laws.

-Ages more specific than whole numbers -Internet protocol (IP) address

-Contact information such as phone number or email address –

Location data

Data that are not directly identifying may also be inappropriate to share, as in combination they can become identifying. For example, data collected from a small group of participants, vulnerable populations, or private groups should not be shared if they involve indirect identifiers (such as sex, ethnicity, location, etc.) that may risk the identification of study participants. Additional guidance on preparing raw data for publication can be found in our Data Policy (https://journals.plos.org/plosone/s/data-availability#loc-human-research-participant-data-and-other-sensitive-data) and in the following article: http://www.bmj.com/content/340/bmj.c181.long. Please remove or anonymize all personal information (<specific identifying information in file to be removed>), ensure that the data shared are in accordance with participant consent, and re-upload a fully anonymized data set. Please note that spreadsheet columns with personal information must be removed and not hidden as all hidden columns will appear in the published file.

• Response: We disagree with this decision. The information included in the Supporting Information Tables S1 and S2 (now Tables S2 and S3) are not identifiable. Kindly explain to us why the editorial staff feels that these tables are not anonymized.

Additional Editor Comments:

The main limitation of the report is that the model used in the study assumes a lifelong duration of vaccine protection and herd immunity which may be unrealistic and unsubstantiated by available data. This assumption potentially overestimates the long-term benefits of vaccination if vaccine-induced immunity wanes over time or if coverage rates fall below expectations.

• Response: We thank the reviewer for their feedback. The assumption of permanent protection given a full-series of HPV vaccine (2-doses for individuals under 15 years of age and 3-doses for those 15 and older) is a widely accepted assumption and has been assumed in most HPV health-economic modeling studies for at least the last 10 years. Recently, statistical studies have shown that among young girls aged 10-14 years receiving 2vHPV vaccination, durability of antibody levels above natural infection level was predicted to be over 70 years for anti-HPV 16 and over 78 years for anti-HPV 18, or even lifelong depending on the model used and even and also on the age of vaccination (see Schwarz et al., 2017 and Schwarz et al., 2019). It is not standard practice to include a sensitivity analysis that assumes waning protection.

Herd immunity is a natural and well understood consequence of vaccination – the degree of that protection is dependent on the model structure, inputs, demographics, and most of the other assumptions that affect the dynamics of disease acquisition. The current model has been in use (in evolving forms) for more than 15 years and has been validated in multiple populations including comparative modeling analyses, and it has been used to directly support national immunization plan policy recommendations in the US (Daniels et al., 2021), UK (Owusu-Edusei et al., 2022), France (Majed et al., 2021), and dozens of other countries around the world (Palmer et al., 2023, Diakite et al., 2023, and Palmer et al., 2024).

As for coverage, we explored different coverage assumptions (varying from 40% to 80%) in the sensitivity analysis.

References:

Schwarz TF, Huang LM, Valencia A, Panzer F, Chiu CH, Decreux A, Poncelet S, Karkada N, Folschweiller N, Lin L, Dubin G, Struyf F. A ten-year study of immunogenicity and safety of the AS04-HPV-16/18 vaccine in adolescent girls aged 10-14 years. Hum Vaccin Immunother. 2019;15(7-8):1970-1979. doi: 10.1080/21645515.2019.1625644. Epub 2019 Jul 17. PMID: 31268383; PMCID: PMC6746471.

Schwarz TF, Galaj A, Spaczynski M, Wysocki J, Kaufmann AM, Poncelet S, Suryakiran PV, Folschweiller N, Thomas F, Lin L, Struyf F. Ten-year immune persistence and safety of the HPV-16/18 AS04-adjuvanted vaccine in females vaccinated at 15-55 years of age. Cance

---

## [Decision Letter · Decision Letter 1]

17 Mar 2025

PONE-D-24-13297R1Health impact and cost-effectiveness analysis of gender-neutral versus female-only 9-valent human papillomavirus vaccination in TaiwanPLOS ONE

Dear Dr. Sukarom,

Thank you for submitting your manuscript to PLOS ONE. After careful consideration, we feel that it has merit but does not fully meet PLOS ONE’s publication criteria as it currently stands. Therefore, we invite you to submit a revised version of the manuscript that addresses the points raised during the review process.

We look forward to receiving your revised manuscript.

Kind regards,

Emmanuel Timmy Donkoh, PhD

Academic Editor

PLOS ONE

Journal Requirements:

Reviewers' comments:

Reviewer's Responses to Questions

**Comments to the Author**

1. If the authors have adequately addressed your comments raised in a previous round of review and you feel that this manuscript is now acceptable for publication, you may indicate that here to bypass the “Comments to the Author” section, enter your conflict of interest statement in the “Confidential to Editor” section, and submit your "Accept" recommendation.

Reviewer #2: All comments have been addressed

Reviewer #3: All comments have been addressed

2. Is the manuscript technically sound, and do the data support the conclusions?

Reviewer #2: Yes

Reviewer #3: Yes

3. Has the statistical analysis been performed appropriately and rigorously? 

Reviewer #2: Yes

Reviewer #3: Yes

4. Have the authors made all data underlying the findings in their manuscript fully available?

Reviewer #2: Yes

Reviewer #3: Yes

5. Is the manuscript presented in an intelligible fashion and written in standard English?

Reviewer #2: Yes

Reviewer #3: Yes

6. Review Comments to the Author

Reviewer #2: The authors have addressed the comments for the initial submission appropriately. There are some new items based on the revision of the manuscript.

1. In Table S1, the authors showed HPV attribution in different HPV-related cancers. The HPV prevalence for penile cancer was stated as 100% in the table, which is unlikely based on the reference that the authors cited. The authors please justify this setting, or revise the value and, if necessary, corresponding results. Also, as the authors said, HPV attribution in cancer cases varies by geographic location and population. Please state whether local data specific to the Taiwanese setting was used or the location that they referred to.

2. The authors presented calibration results to cancer incidence that was related to 4vHPV and 9vHPV. The authors please state whether calibration has been done to the incidence that was related to non-vaccine-targeted HPV types, and how the model handles the disease burden that is related to non-vaccine-targeted HPV types.

3. The authors calibrated the model to the age-standardized rate of cancer incidence. The age-specific incidence rate, e.g., the peak and the trend, varies by age and cancer type. The authors please state the potential impact of calibrating the model outputs to an age-standardized rate instead of an age-specific rate, e.g., this approach affects the projected incidence and so the associated costs and health benefits across time horizon.

Reviewer #3: Thank you for the opportunity to review this manuscript. It provides additional information on the optimal program structure around HPV vaccination, to hasten elimination of HPV-related diseases across the world. Since I have noted the questions from the previous set of reviews and the detailed responses from the authors, I would like to focus on a more policy/implementation perspective; a policy-level 'so what' as it were.

1. Most health systems across the world are struggling with sustainable financing, in the setting of ageing populations as well as other remerging threats. With the exception of genital warts, the health impact of on the other conditions is reduced by both fewer numbers, coupled with relatively low incidence of some of the conditions (like respiratory papillomatosis). What would be the implications for decision-makers, who have to consider many factors before investing in gender neutral HPV vaccination?

2. Most of the health impact and cost savings from averted managements costs emanate from conditions associated with HPV 6, 11, 16 and 18. Though this may not be relevant to Taiwan that already is implementing the 9vHPV, what can these finding inform policymakers, especially in LMICs, considering whether to 9vHPV in their national programs?

3. These findings are Taiwan-specific and fit into the health system realties of this setting. However, by publishing, the authors also aim to guide decision-making and policy initiatives across the world. Looking at the relative burden of various HPV-related conditions across the world, what would the authors recommend to countries oaf various economic contexts, based on the insights from their study? Focus on one-dose strategies but with high coverage among girls? Gender-neutral one-dose strategy? Girls-only two doses? While the current model did not address all these scenarios together (just compared FOV vs GNV), the authors can bring together other modelling studies published in the last few years to weave a narrative that can guide program managers and teams across the world.

4. It would also add value even to the study context, to summarize what the total cost implication would be, if Taiwan was to plan to intrude gender neutral HPV vaccine, in relation to its current health expenditure (more broadly or more specifically on vaccination). This can then be extrapolated to the investment needed to get the benefits within the time horizon specified. It is important to note that this information is already available within the CEA determinations, but for policy-level decision-making, cost implication is vital for planning.

5. As part of the responses to a previous review, the authors stated that since Taiwan is not considering a single-done program, there was no need to mention it in their discussion, a position I disagree with. We publish to share our findings, put them in context, and also discuss their generalizability and implications to global audiences. It would value for this study to contribute to the discussion that many countries are having around gender neutral, extended catch-up vaccination, one-dose strategies, etc.

My points above are made with the understanding that some of these points may not have been addressed by the current model, but nevertheless warrant being part of the discussion and recommendation.

7. PLOS authors have the option to publish the peer review history of their article (what does this mean?). If published, this will include your full peer review and any attached files.

Reviewer #2: No

Reviewer #3: **Yes: **Valerian Mwenda

---

## [Author Response · Author response to Decision Letter 2]

9 Jul 2025

July 9, 2025

Emmanuel Timmy Donkoh, PhD

Academic Editor

PLOS ONE

Dear Dr. Emmanuel Timmy Donkoh,

On behalf of my co-authors, I would like to thank you and the reviewers for your feedback on our submission entitled, “Health impact and cost-effectiveness analysis of gender-neutral versus female-only 9-valent human papillomavirus vaccination in Taiwan” (PONE-D-24-13297). Below please find our point-by-point responses to the comments raised during peer review.

Reviewer #2:

The authors have addressed the comments for the initial submission appropriately. There are some new items based on the revision of the manuscript.

1. In Table S1, the authors showed HPV attribution in different HPV-related cancers. The HPV prevalence for penile cancer was stated as 100% in the table, which is unlikely based on the reference that the authors cited. The authors please justify this setting, or revise the value and, if necessary, corresponding results. Also, as the authors said, HPV attribution in cancer cases varies by geographic location and population. Please state whether local data specific to the Taiwanese setting was used or the location that they referred to.

• Response: We have reviewed and revised the HPV prevalence data. At the time the analyses were conducted, the input data had been updated multiple times, resulting in several versions of the input sheets. In the initial submission, we inadvertently included an outdated version of the table. We have now corrected this and provided the updated S1 table. The S1 table also includes references indicating which data are specific to Taiwan and which were sourced from other countries due to the lack of local data. All analyses have been re-run using the correct, updated inputs.

2. The authors presented calibration results to cancer incidence that was related to 4vHPV and 9vHPV. The authors please state whether calibration has been done to the incidence that was related to non-vaccine-targeted HPV types, and how the model handles the disease burden that is related to non-vaccine-targeted HPV types.

• Response: The current manuscript only focused on the cancer incidences that was related to 4vHPV and 9vHPV, proportionated based on attribution fraction of each vaccine-targeted HPV genotypes, thereby focusing on the direct impact of vaccination.

3. The authors calibrated the model to the age-standardized rate of cancer incidence. The age-specific incidence rate, e.g., the peak and the trend, varies by age and cancer type. The authors please state the potential impact of calibrating the model outputs to an age-standardized rate instead of an age-specific rate, e.g., this approach affects the projected incidence and so the associated costs and health benefits across time horizon.

• Response: Calibrating model outputs to the age-standardized rate (ASR) of cancer incidence, instead of using age-specific incidence rates, may simplify the analysis but it may also lead to loss of granularity of the data and thereby affecting the projected incidence, associated healthcare cost and health benefits across different age cohorts over time. For example, ASR allows for a more uniform comparison across different populations and time periods by removing the effects of differing age distributions. This can lead to a more accurate understanding of trends in cancer incidence that are not influenced by demographic shifts. However, it could mask important trends in specific age groups that may indicate increasing or decreasing risks associated with certain cancers, which in turn may under or overestimate healthcare costs. That is, if an age-specific rate indicates a rising incidence in older adults, cost projections tied to treatment and care for this demographic could be higher than if only ASRs are considered. Conversely, focusing solely on ASRs might overlook increasing needs for younger populations facing rising incidences of certain cancers. Similar implications can be delineated for health benefit assessments, policy and resource allocations. ASR might obscure these nuances, especially if certain age cohorts are increasingly diagnosed with specific types of cancer. We have included a discussion of this limitation in the manuscript.

Reviewer #3:

Thank you for the opportunity to review this manuscript. It provides additional information on the optimal program structure around HPV vaccination, to hasten elimination of HPV-related diseases across the world. Since I have noted the questions from the previous set of reviews and the detailed responses from the authors, I would like to focus on a more policy/implementation perspective; a policy-level 'so what' as it were.

1. Most health systems across the world are struggling with sustainable financing, in the setting of ageing populations as well as other remerging threats. With the exception of genital warts, the health impact of on the other conditions is reduced by both fewer numbers, coupled with relatively low incidence of some of the conditions (like respiratory papillomatosis). What would be the implications for decision-makers, who have to consider many factors before investing in gender neutral HPV vaccination?

• Response: Thank you for your insightful question. Decision-makers must weigh multiple factors when considering investments in gender-neutral HPV vaccination, particularly in the context of constrained healthcare financing. While some HPV-related conditions have lower incidence, gender-neutral vaccination provides broader herd immunity, reducing overall transmission and long-term healthcare costs. Additionally, it helps protect against conditions with significant morbidity, such as head and neck cancers (including oropharyngeal cancers), which are rising in incidence and have substantial treatment burdens. A previous publication by Man et al., (Building resilient cervical cancer prevention through gender-neutral HPV vaccination. Elife. 2023 Jul 24;12:e85735. doi: 10.7554/eLife.85735. PMID: 37486822; PMCID: PMC10365835) emphasizes that gender-neutral HPV vaccination is an effective strategy to improve the resilience to disruption of cancer prevention programs and to enhance the progress towards cervical cancer elimination.

A comprehensive cost-effectiveness analysis, considering both direct and indirect benefits, is essential for informed decision-making.

2. Most of the health impact and cost savings from averted managements costs emanate from conditions associated with HPV 6, 11, 16 and 18. Though this may not be relevant to Taiwan that already is implementing the 9vHPV, what can these finding inform policymakers, especially in LMICs, considering whether to 9vHPV in their national programs?

• Response: Thank you for your question. While Taiwan has already implemented the 9vHPV, these findings are particularly relevant for policymakers in LMICs considering its adoption. The majority of health impact and cost savings stem from preventing conditions caused by HPV 6, 11, 16, and 18, which contribute significantly to both morbidity and treatment costs. The following text has been added to the Discussion section: “In the present study, the 9vHPV GNV strategy resulted in savings of NTD 1,574,288,155 in disease management costs compared to the 9vHPV FOV strategy. These cost reductions were attributable to diseases associated with HPV 16/18 (43.0%), HPV 6/11 (46.4%), and HPV 31/33/45/52/58 (10.6%). Notably, the incremental cost of implementing GNV—estimated at NTD 358.23 per person in the base case—can be contextualized as a percentage of the current cervical cancer screening budget (NTD 941 million) or the broader cancer prevention budget (NTD 3.398 billion), offering policymakers a clearer picture of the investment-to-benefit ratio [65]. These cost-reduction results align with those found in a modeling exercise in Hong Kong, where cost reductions attributable to HPV-related diseases were 52.1% for HPV16/18, 41.4% for HPV6/11, and 6.6% for HPV 31/33/45/52/58 [64]. These findings may be particularly valuable for other low- and middle-income countries considering the adoption of a 9vHPV GNV strategy where budgetary constraints often limit access to broader HPV vaccination programs, and the burden of HPV-related diseases remains significant.”

3. These findings are Taiwan-specific and fit into the health system realties of this setting. However, by publishing, the authors also aim to guide decision-making and policy initiatives across the world. Looking at the relative burden of various HPV-related conditions across the world, what would the authors recommend to countries oaf various economic contexts, based on the insights from their study? Focus on one-dose strategies but with high coverage among girls? Gender-neutral one-dose strategy? Girls-only two doses? While the current model did not address all these scenarios together (just compared FOV vs GNV), the authors can bring together other modelling studies published in the last few years to weave a narrative that can guide program managers and teams across the world.

• Response: Based on the findings from our study, we support the adoption of a gender-neutral 2-dose HPV vaccination strategy, which has also been shown to be cost-effective in other settings, including Hong Kong.

In this analysis, we modeled a 2-dose vaccination approach consistent with Taiwan’s current national recommendation. While we acknowledge that countries differ in healthcare infrastructure, program history, and implementation capacity—and may therefore consider alternative strategies—our analysis is specific to the context of Taiwan, where the HPV vaccination program has been in place for less than 10 years and has primarily focused on females.

4. LIt would also add value even to the study context, to summarize what the total cost implication would be, if Taiwan was to plan to intrude gender neutral HPV vaccine, in relation to its current health expenditure (more broadly or more specifically on vaccination). This can then be extrapolated to the investment needed to get the benefits within the time horizon specified. It is important to note that this information is already available within the CEA determinations, but for policy-level decision-making, cost implication is vital for planning.

• Response: Thank you for this valuable suggestion. We agree that providing the total cost implications of adopting a gender-neutral HPV vaccination strategy would enhance the relevance of our findings for policy-level decision-making.

Based on our cost-effectiveness analysis, the incremental cost of implementing a gender-neutral 2-dose HPV vaccination program—relative to the current female-only program—can be framed in the context of Taiwan’s current health expenditure. While the HPV vaccine budget is managed separately from other public vaccine programs and not included in the general vaccine budget, it is still helpful to understand the broader fiscal landscape. In 2024, the total public vaccine budget is estimated at 12.683 billion New Taiwan Dollars (NTD), although this does not include the HPV vaccine program specifically [See: 4382be02-6f5e-41c3-98e9-d975675824e1.pdf]. Additionally, the cancer prevention budget is projected at 3.398 billion NTD, with cervical cancer screening accounting for 941 million NTD [See: https://dep.mohw.gov.tw/DOA/cp-650-79793-112.html]. Given the relatively modest incremental investment required for gender-neutral vaccination—when compared to these broader budget figures—this strategy could represent a financially feasible option. Furthermore, the long-term reductions in HPV-related diseases and treatment costs would contribute to improved efficiency and sustainability of cancer prevention efforts. For example, the cost of implementing gender-neutral vaccination over the modeled time horizon could be contextualized as a percentage of the current cervical cancer screening or cancer prevention budget, offering policymakers a clearer picture of the investment-to-benefit ratio.

We have added this information to the Discussion section, stating: “Notably, the incremental cost of implementing GNV—estimated at NTD 358.23 per person in the base case—can be contextualized as a percentage of the current cervical cancer screening budget (NTD 941 million) or the broader cancer prevention budget (NTD 3.398 billion), offering policymakers a clearer picture of the investment-to-benefit ratio [65].”

As part of the responses to a previous review, the authors stated that since Taiwan is not considering a single-done program, there was no need to mention it in their discussion, a position I disagree with. We publish to share our findings, put them in context, and also discuss their generalizability and implications to global audiences. It would value for this study to contribute to the discussion that many countries are having around gender neutral, extended catch-up vaccination, one-dose strategies, etc.

• Response: A thorough discussion of 1-dose vs. 2-dose HPV vaccination strategies has been added to the Discussion section. The updated text reads: “In Taiwan, the National Immunization Program recommendation under consideration is for full-series, two-dose, on-label use of the HPV vaccine for the cohorts included in this study. As such, our analyses focused specifically on the two-dose regimen and did not consider one-dose vaccination strategies, as they fall outside the scope of Taiwan’s current vaccination strategy. Although emerging evidence suggests the potential 10-year effectiveness of single-dose HPV vaccination, there remains uncertainty regarding its public health impact, namely on cancers and diseases prevented. For example, a cost-effectiveness analysis by Daniels et al. (2022) found a significant increase in HPV-related cancer cases over a 100-year period when comparing a one-dose versus two-dose vaccination program in the United Kingdom [62]. Furthermore, the one-dose vaccination program was also found to be less cost-effective than the two-dose vaccination program [62]. Similar conclusions have also been observed in the context of low-income countries such as Indonesia in which two-dose HPV vaccination programs are found to be more cost-effective compared to one-dose programs across various scenarios [63].”

My points above are made with the understanding that some of these points may not have been addressed by the current model, but nevertheless warrant being part of the discussion and recommendation.

---

## [Decision Letter · Decision Letter 2]

19 Sep 2025

Health impact and cost-effectiveness analysis of gender-neutral versus female-only 9-valent human papillomavirus vaccination in Taiwan

PONE-D-24-13297R2

Dear Dr. Sukarom,

We’re pleased to inform you that your manuscript has been judged scientifically suitable for publication and will be formally accepted for publication once it meets all outstanding technical requirements.

Kind regards,

Emmanuel Timmy Donkoh, PhD

Academic Editor

PLOS ONE

Reviewers' comments:

Reviewer's Responses to Questions

**Comments to the Author**

1. If the authors have adequately addressed your comments raised in a previous round of review and you feel that this manuscript is now acceptable for publication, you may indicate that here to bypass the “Comments to the Author” section, enter your conflict of interest statement in the “Confidential to Editor” section, and submit your "Accept" recommendation.

Reviewer #2: All comments have been addressed

Reviewer #3: All comments have been addressed

2. Is the manuscript technically sound, and do the data support the conclusions?

Reviewer #2: Yes

Reviewer #3: Yes

3. Has the statistical analysis been performed appropriately and rigorously? 

Reviewer #2: Yes

Reviewer #3: Yes

4. Have the authors made all data underlying the findings in their manuscript fully available?

Reviewer #2: Yes

Reviewer #3: Yes

5. Is the manuscript presented in an intelligible fashion and written in standard English?

Reviewer #2: Yes

Reviewer #3: Yes

6. Review Comments to the Author

Reviewer #2: The authors have addressed the comments in the previous round of review.

Reviewer #3: I feel the authors' responses are adequate and satisfactory and have made the required revisions on the manuscript.

7. PLOS authors have the option to publish the peer review history of their article (what does this mean?). If published, this will include your full peer review and any attached files.

Reviewer #2: No

Reviewer #3: **Yes: **Valerian Mwenda

---

## [Editor Report · Acceptance letter]

PONE-D-24-13297R2

PLOS ONE

Dear Dr. Sukarom,

I'm pleased to inform you that your manuscript has been deemed suitable for publication in PLOS ONE. Congratulations! Your manuscript is now being handed over to our production team.

Kind regards,

on behalf of

Dr. Emmanuel Timmy Donkoh

Academic Editor

PLOS ONE